# Guided-Motion Bicruciate-Stabilized Total Knee Arthroplasty Reproduces Native Medial Collateral Ligament Strain

**DOI:** 10.3390/medicina58121751

**Published:** 2022-11-29

**Authors:** Dai-Soon Kwak, Yong Deok Kim, Nicole Cho, Ho-Jung Cho, Jaeryong Ko, Minji Kim, Jae Hyuk Choi, Dohyung Lim, In Jun Koh

**Affiliations:** 1Catholic Institute for Applied Anatomy, Department of Anatomy, College of Medicine, The Catholic University of Korea, Seoul 06591, Republic of Korea; 2Joint Replacement Center, Eunpyeong St. Mary’s Hospital, Seoul 03312, Republic of Korea; 3Department of Orthopaedic Surgery, College of Medicine, The Catholic University of Korea, Seoul 06591, Republic of Korea; 4Boston College, Morrissey College of Arts and Sciences, Chestnut Hill, MA 02467, USA; 5Department of Mechanical Engineering, Sejong University, Seoul 05006, Republic of Korea

**Keywords:** knee, medial collateral ligament, strain, video extensometer, total knee arthroplasty

## Abstract

*Background and Objectives*: Guided-motion bicruciate-stabilized (BCS) total knee arthroplasty (TKA) includes a dual cam-post mechanism with an asymmetric bearing geometry that promotes normal knee kinematics and enhances anterior-posterior stability. However, it is unclear whether the improved biomechanics after guided-motion BCS TKA reproduce soft tissue strain similar to the strain generated by native knees. The purpose of this cadaveric study was to compare medial collateral ligament (MCL) strain between native and guided-motion BCS TKA knees using a video extensometer. *Materials and Methods*: Eight cadaver knees were mounted onto a customized knee squatting simulator to measure MCL strain during flexion in both native and guided-motion BCS TKA knees (Journey II-BCS; Smith & Nephew, Memphis, TN, USA). MCL strain was measured using a video extensometer (Mercury^®^ RT RealTime tracking system, Sobriety s.r.o, Kuřim, Czech Republic). MCL strain level and strain distribution during knee flexion were compared between the native and guided-motion BCS TKA conditions. *Results*: The mean and peak MCL strain were similar between native and guided-motion BCS TKA knees at all flexion angles (*p* > 0.1). MCL strain distribution was similar between native and BCS TKA knees at 8 of 9 regions of interest (ROIs), while higher MCL strain was observed after BCS TKA than in the native knee at 1 ROI in the mid portion of the MCL at early flexion angles (*p* < 0.05 at ≤30° of flexion). *Conclusions:* Guided-motion BCS TKA restored the amount and distribution of MCL strain to the values observed on native knees.

## 1. Introduction

Despite advancements in total knee arthroplasty (TKA), many patients have residual symptoms or unsatisfactory outcomes after TKA [1,2,3,4,5,6]. Recently, guided-motion bicruciate-stabilized (BCS) TKA, which involves asymmetric bearing geometry and dual substitution for the anterior cruciate ligament (ACL) and posterior cruciate ligament, has been introduced [7,8,9]. Guided-motion BCS TKA has an asymmetric femoral component, a polyethylene insert with 3° of tibial varus, a medially concave and laterally convex shape, and a dual cam-post mechanism (Figure 1). The goal of guided-motion BCS TKA is to facilitate guided motion that is closest to normal knee kinematics. A growing body of evidence supports that guided-motion BCS TKA can mimic native knee kinematics, improve recovery and activity, and provide more natural knee sensations compared to conventional TKA [5,6,8,9,10,11]. However, the biomechanical mechanisms that underlie these improved results remain unclear.

As the medial collateral ligament (MCL) plays critical roles in primary restraint against mechanical stresses and neurosensory feedback, changes in MCL tension after TKA affect postoperative kinematics and outcomes. Therefore, restoration of appropriate MCL strain is essential for optimal performance after TKA. Despite the significant influence of the ACL on restoring native knee kinematics, most TKA designs sacrifice the ACL without providing a substitute for its function. In the conventional ACL-deficient TKA design, the MCL and remaining capsular structures provide restraint against anterior tibial translation. Previous studies have reported that MCL strain, compared to native knees, worsened after conventional ACL-deficient TKA [12,13,14,15,16,17,18]. Theoretically, BCS TKA may restore post-TKA MCL strain to levels observed in native knees by enhancing anterior-posterior stability. However, no previous studies have investigated the changes in MCL strain after BCS TKA, and it is unclear whether BCS TKA restores native strain of the MCL.

The purpose of this study was to compare MCL strain between native knees and guided-motion BCS TKA knees using a video extensometer, a highly accurate tool for measuring surface strain in human cadavers [19,20]. We hypothesized that guided-motion BCS TKA would restore the strain level and distribution of the MCL to the values observed in native healthy knees. 

## 2. Materials and Methods

Eight fresh-frozen knees from eight men with a mean age of 79 years (range: 49–96 years) were used. The specimens were macroscopically intact and did not exhibit any gross pathology. The specimens were frozen at −20 °C until the evening before dissection, when they were thawed at room temperature. The lower extremity specimens were prepared by disarticulating the hip joint; high-resolution photographs were obtained to measure the preoperative hip-knee-ankle axis. The hip-knee-ankle axis showed a varus angle of 1.3 ± 3.2° (range: varus 4° to valgus 6°). The skin and subcutaneous tissue were dissected without damage to the extensor mechanism, retinaculum, knee capsule, or periarticular soft tissues. The quadriceps femoris was separated into the vastus medialis, rectus femoris/vastus intermedius, and vastus lateralis. The hamstring muscles were separated into the biceps femoris and semimembranosus/semitendinosus. The separated muscle branches were sutured using wire to connect the muscles and allow transmission of force. The femur was cut 30 cm proximal to the joint line; the tibia was cut 25 cm distal to the joint line. The ends of the femur and tibia were anatomically positioned and a cylindrical resin mold (Z-Grip, Evercoat, OH, USA) was created to mount the knee squatting simulator. Each specimen was securely mounted in its original axial alignment on a customized knee squatting simulator system (RNX & Corentec, Seoul, Republic of Korea) based on the original Oxford rig (Figure 2) [21]. This system, which is an opened loop control system with a motor type actuator with 7.5º/sec angular velocity, produced continuous flexion-extension motion while permitting physiological muscle loading and 6° of freedom positioning. The multiplane loading of the quadriceps and hamstring muscles was used to simulate physiological loading of the knee joint (vastus medialis 51 N, rectus femoris/ vastus intermedius 87 N, and vastus lateralis 77 N, biceps femoris 31 N, semimembranosus/ semitendinosus 54 N). The total loading was 300 N [17]. These loading parameters were based on the ratios of the physiological cross-sectional area of the muscles as described in a previous anatomical study [22]. The muscles were loaded in a multiplanar fashion as described in previous studies [23,24].

A single experienced surgeon (one of authors) performed the arthroplasties using the JOURNEYTM II Bi-cruciate Stabilized Total Knee System (Smith & Nephew, Memphis, TN, USA). A subvastus approach was used to expose the knee joint, whereas the patellae were left un-resurfaced. TKA was performed using the conventional measured resection technique. The distal femur was resected using individualized intramedullary instrumentation based on the difference between the mechanical and anatomical axes of the specimen. The trans-epicondylar axis served as a reference to determine the extent of external rotation of the femoral component. Coronal and sagittal resection of the proximal tibia was performed using extramedullary instrumentation at a cutting angle of 90° to the tibial axis. Finally, flexion gaps at 0 and 90° were measured using a tensor device (B Braun-Aesculap, Tuttlingen, Germany) with a 200-N distraction force [25]. 

The superficial MCL was identified as previously described, then stained with multiple random speckles prior to measurement of MCL strain [18,26]. Real-time changes in MCL strain during flexion were analyzed using a non-contact video extensometer that consisted of a high-resolution digital camera (ISG, MONET 3D, Sobriety s.r.o, Czech Republic) and real-time image processing software (ISG, Mercury RT x64 2.7, Sobriety s.r.o). The camera was installed 1 m from the specimen to produce a field of view of 485 (w) × 383 mm and resolution of 1.87 µm. Illumination was minimized by performing the tests in darkness under two 36-watt light-emitting diode lights (Figure 3). MCL strain was measured at knee flexion angles of 0–120°, at intervals of 15°. Repeatability was checked in real-time to raise reliability. Each measurement was performed in three flexion-extension cycles and all of graphs following each of the three cycles were plotted at the same time. The patterns of all three cycles were compared and repeatability was analyzed. After repeatability was confirmed, data from the 3rd cycle were used for statistical analyses in all specimens. The native and post-TKA MCL strains were compared in terms of quantity and distribution. The mean MCL strain was determined by measuring the strain over the entire MCL and the peak MCL strain at one-fourth of the entire MCL area. The distribution of MCL strain was evaluated by measuring MCL strain at 9 regions of interest (ROIs), drawn by dividing the ligament vertically (front to back) into equal anterior (A), middle (M), and posterior (P) regions, and horizontally (top to bottom) into equal regions numbered 1–3. MCL strain was measured in all specimens before and after arthroplasty.

### Statistical Analysis

Data are presented as means ± standard deviations. Paired t-tests were used to determine whether the mean and peak MCL strains differed between native and post-TKA knee specimens. The Shapiro–Wilk test confirmed that this data set was normally distributed. The variables subjected to multiple between-group comparisons included the mean, peak, and distribution of MCL strain, which were analyzed using repeated-measures ANOVA, followed by the Bonferroni corrected post hoc test in order to protect against multiple comparison bias. In addition, paired t-tests were used to determine differences in the distribution of MCL strain at 9 ROIs between native and BCS TKA knee specimens. Data analysis was performed using SPSS software for Windows (ver. 26.0; IBM Corp., Armonk, NY, USA). *p* < 0.05 was considered to indicate statistical significance. A priori power analysis based on the results of our previous study of changes in the MCL strain in native knees was performed to determine the necessary sample size to achieve sufficient statistical power. Using a two-sided hypothesis test at an alpha level of 0.05 and power of 80%, 7 knees were required to detect a 5% difference. A 5% change was considered biomechanically meaningful since ligament damage occurs at 5% strain [27].

## 3. Results

Guided-motion BCS TKA restored the mean and peak MCL strain to the levels present in native knees during knee flexion. Guided-motion BCS TKA provided mean strain measurements similar to the levels in native knees at all flexion angles [Mean (SD) native vs. J2 BCS mean MCL strain (%) at 15°, 2.4 (1.5) vs. 3.5 (1.1), *p* = 0.96; at 30°, 3.7 (2.7) vs. 5.2 (2.4), *p* = 0.43; at 45°, 4.5 (3.0) vs. 6.0 (3.0), *p* = 0.48; at 60°, 5.1 (3.1) vs. 6.6 (3.7), *p* = 0.56; at 75°, 5.4 (3.4) vs. 7.1 (4.4), *p* = 0.64; at 90°, 5.6 (3.5) vs. 7.6 (5.2), *p* = 0.72; at 105°, 5.9 (3.7) vs. 8.0 (5.7), *p* = 0.88; at 120°, 5.9 (3.9) vs. 8.2 (5.9), *p* = 0.80] (Figure 4). The peak strains at one-fourth of the MCL area were similar between guided-motion BCS TKA and native knees [Mean (SD) native vs. J2 BCS peak MCL strain (%) at 15°, 5.3 (4.3) vs. 7.7 (3.4), *p* = 0.32; at 30°, 11.5 (10.9) vs. 14.3 (13.9), *p* = 0.87; at 45°, 14.6 (11.1) vs. 17.2 (16.7), *p* = 0.92; at 60°, 17.3 (11.4) vs. 19.1 (17.3), *p* = 0.96; at 75°, 18.6 (12.0) vs. 20.6 (17.5), *p* = 0.99; at 90°, 19.4 (12.5) vs. 21.3 (17.4), *p* = 0.93; at 105°, 20.3 (12.3) vs. 21.4 (16.4), *p* = 0.70; at 120°, 20.6 (12.0) vs. 20.8 (14.2), *p* = 0.94] (Figure 5). 

Guided-motion BCS TKA restored the native knee MCL strain distribution. Significant strain differences between guided-motion BCS TKA and native knees were observed in 1 of the 9 ROIs (i.e., at M2). In this ROI, strain was higher after guided-motion BCS TKA than in native knees only at ≤30° of flexion [Mean (SD) native vs. J2 BCS MCL strain at M2 (%) at 15°, 0.69 (0.21) vs. 2.4 (0.4), *p* = 0.048; at 30°, 0.74 (0.30) vs. 2.9 (0.6), *p* = 0.024; at 45°, 1.3 (0.4) vs. 3.2 (0.7), *p* = 0.12; at 60°, 1.7 (0.5) vs. 3.4 (0.9), *p* = 0.32; at 75°, 1.9 (0.6) vs. 3.7 (1.2), *p* = 0.72; at 90°, 2.0 (0.7) vs. 3.9 (1.5), *p* = 0.93; at 105°, 2.0 (0.7) vs. 4.3 (1.9), *p* = 0.99; at 120°, 1.9 (0.7) vs. 4.6 (2.0), *p* = 0.94] (Figure 6).

## 4. Discussion

Despite improvements in TKA design, most TKA systems sacrifice the ACL, which may lead to suboptimal satisfaction and function. Guided-motion BCS TKA replicates the normal knee motion by promoting kinematics and anterior-posterior stability. Enhanced stability after guided-motion BCS TKA may normalize the soft tissue tension that contributes the perception of a normal knee. In this cadaveric study, we compared MCL strain between native and post-BCS TKA knees using a video extensometer to determine whether guided-motion BCS TKA restores MCL strain to the level found in native knees. 

Our findings suggest that guided-motion BCS TKA normalizes MCL strain. In this study, no significant differences were observed in the mean and peak strain measurements between native and guided-motion BCS-TKA knees at any flexion angle. The MCL provides restraint against anterior tibial translation in ACL-deficient knees [28]. Multiple previous studies have reported that MCL strain was significantly higher in post-TKA knees than in native knees [14,15,18] and postoperative MCL laxity was also higher in post-TKA knees than in native knees [16,17]. Our findings, when taken into account with previous studies, suggest that guided-motion BCS TKA successfully restored MCL strain to the level found in native knees. Additionally, free nerve endings that serve as a nociceptive system were reported to be the most commonly observed mechanoreceptors in the MCL [29]. Our findings, when taken into account with this anatomical detail, suggest that guided-motion BCS TKA may provide more normal feelings of the knee, which is strongly associated with patient satisfaction [11,12,30]. However, future studies that evaluate the MCL strain thresholds necessary to perceive the differences between normal and prosthetic knees are needed. Our findings also indicated that the standard deviations of the peak MCL strain were much higher than those of the mean MCL strain. One plausible explanation is that the difference in measuring area may affect MCL strain. In this study, the mean MCL strain was determined by measuring the strain over the whole MCL area and the peak MCL strain was found at one-fourth of the entire MCL area, and the distribution of MCL strain was evaluated at each of ROIs. Therefore, the surface strains measured at a smaller area, such as the peak MCL strain and strain at ROIs, are more susceptible to specimen-specific anatomical conditions, such as the bone contour underneath the measured MCL area and the soft tissues connected soft tissues to the MCL, than those measured at the entire MCL.

The results of the present study suggest that guided-motion BCS TKA can normalize the MCL strain distribution pattern. We found that guided-motion BCS TKA restored the strain to the level generated in native knee at 8 of 9 ROIs. Higher MCL strain was found at 1 ROI in the mid-portion of the MCL during flexion at ≤30°. A previous cadaveric study discovered significantly higher strain after conventional ACL-deficient mechanically aligned TKA, compared to native knees, at all flexion angles in the proximal mid to posterior portion of the MCL [18]. Similar to the findings of previous studies, the current results suggest that guided-motion BCS TKA was able to better restore the native MCL strain pattern than conventional TKA. Considering the role of the MCL in neurosensory feedback, patients undergoing BCS TKA may experience greater perception of a normal knee, compared to patients who undergo conventional TKA. Yet, further studies evaluating the relationship between MCL strain and neurosensory feedback changes are required.

There were several noteworthy limitations in this cadaveric study. First, the specimen preparation and testing environment may not have replicated natural conditions and normal squatting loads and patterns. Properties of the MCL may have been changed following rigorous soft tissue balancing, which could have affected the measured surface strain. Additionally, we used muscle loads from previous cadaveric studies investigating knee squatting motions, to simulated physiological loads [23,24], but these loading parameters were too small to simulate in a normal-weight person. This should be considered before extrapolating our cadaveric test findings to real clinical scenarios. Second, we only tested guided-motion BCS TKA implants without comparing them to conventional ACL-deficient TKA implants. Thus, we compared our data with previous studies that investigated conventional mechanically aligned TKA. Third, it is difficult to determine the clinical relevance of our findings because both the threshold of MCL strain that triggers nociception and the cut-off strain value for mechanical failure in the human knee are unknown. Finally, video extensometer analyses have some inherent limitations in terms of image processing, such as poor resolution and quality, as well as distortion of the digital image. Nevertheless, this novel, non-contact analysis technique eliminates the need for strain gauge implantation, thereby reducing the risk of changes to ligament properties. Despite its limitations, this cadaveric study is the first to report MCL strain measurements using a video extensometer after guided-motion BCS TKA. Therefore, our results provide valuable information regarding MCL strain patterns following guided-motion BCS TKA.

## 5. Conclusions

Our study demonstrates that guided-motion BCS TKA restores the amount and distribution of MCL strain to the levels found in native knees during knee flexion. These findings may explain the perception of a normal knee after guided-motion BCS TKA, rather than conventional ACL-deficient TKA.

## Figures and Tables

**Figure 1 medicina-58-01751-f001:**
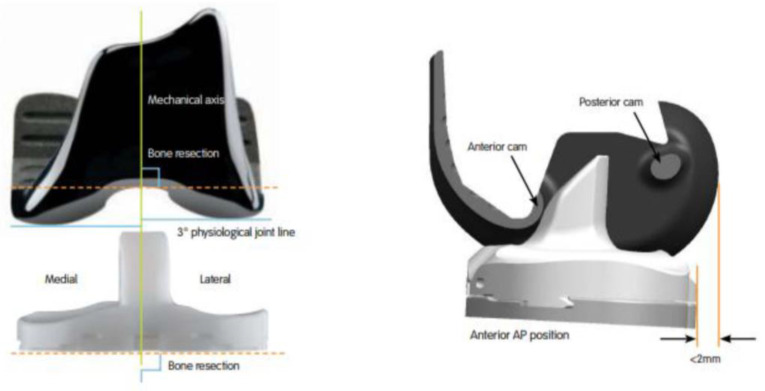
The JOURNEYTM II Bi-cruciate Stabilized Total Knee System (Smith & Nephew, Memphis, TN, USA) has an asymmetric femoral component, a polyethylene insert replicating 3° of the tibial varus, a medially concave and laterally convex shape, and a dual cam-post mechanism.

**Figure 2 medicina-58-01751-f002:**
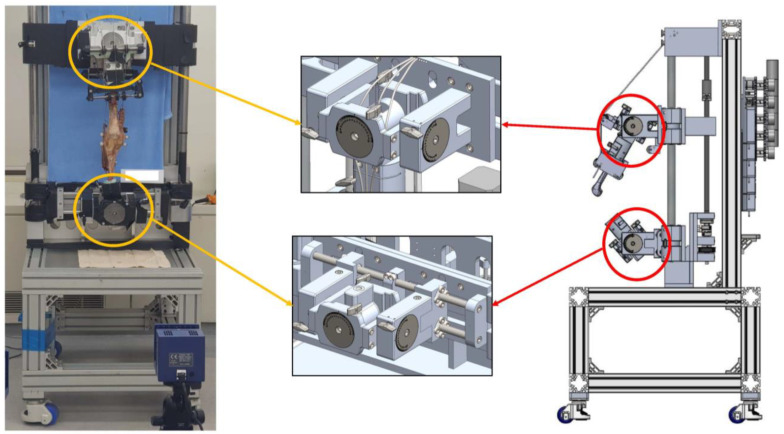
Schematic drawing of knee squatting simulator with six degrees-of-freedom.

**Figure 3 medicina-58-01751-f003:**
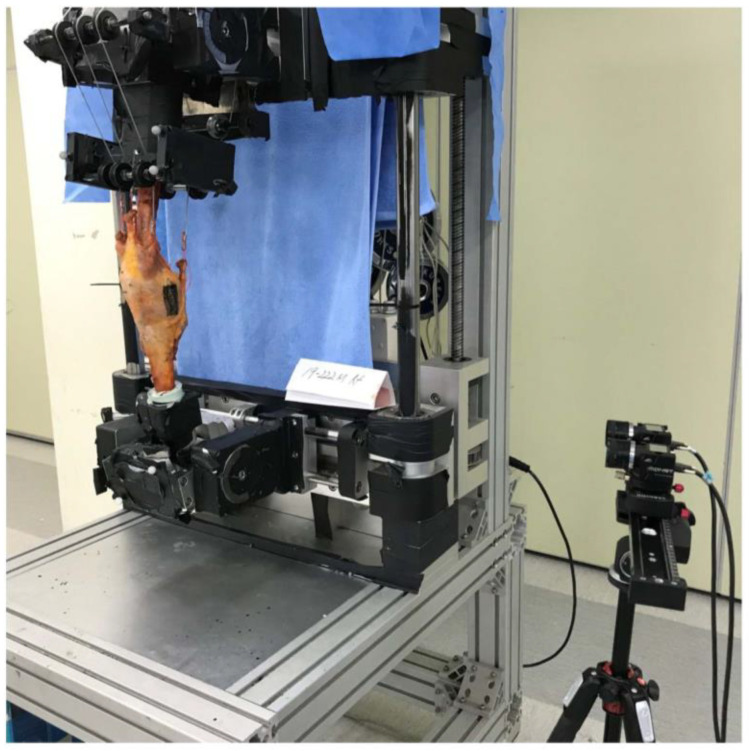
Experimental setup for biomechanical testing. A high-resolution digital camera was installed 1 m from the specimen, which was mounted onto a customized knee squatting simulator system.

**Figure 4 medicina-58-01751-f004:**
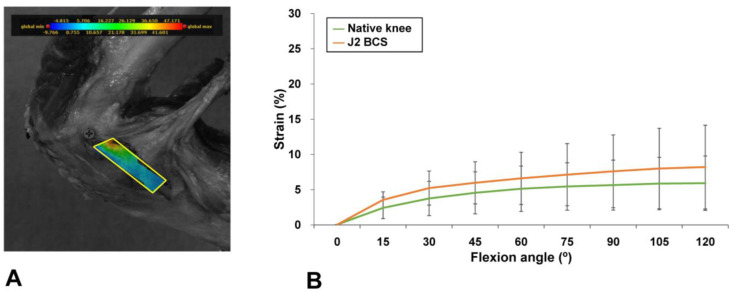
Area for mean MCL strain measurements (**A**) and comparisons of mean MCL strain between native and BCS TKA knees (**B**). The mean strain after BCS TKA was similar to the mean strain in native knees at all flexion angles. Error bars indicate standard deviations. The correspondence between color and strain is shown in the color bar.

**Figure 5 medicina-58-01751-f005:**
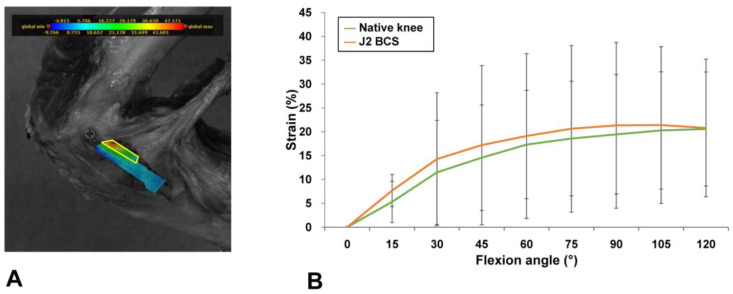
Area for peak MCL strain measurements (**A**) and comparisons of peak MCL strain between native and BCS TKA knees (**B**). The peak strain was measured at one-fourth of the entire MCL area, including the highest MCL strain portion. The peak MCL strain after BCS TKA was similar to the peak MCL strain of native knees at all flexion angles. Error bars indicate standard deviations. The correspondence between color and strain is shown in the color bar.

**Figure 6 medicina-58-01751-f006:**
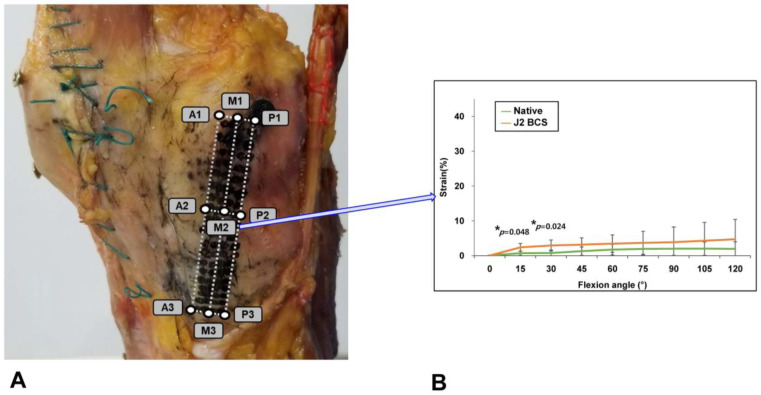
MCL strain distribution. Division of the MCL into 9 regions of interest (ROIs). The MCL was divided vertically into three segments labeled from front to back as anterior (A), middle (M), and posterior (P), and horizontally into three regions numbered 1–3 from top to bottom (**A**). Similar MCL strain was observed at 8 of 9 ROIs at all flexion angles; higher MCL strain after BCS TKA was observed at region M2 during early flexion, compared to native knees. (**B**) Error bars indicate standard deviations. Significant differences (*p* < 0.05) are marked with asterisks.

## Data Availability

All data are presented in the article. Instrumental readings are available upon request from the corresponding author.

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
