# Peer review of "Guided-Motion Bicruciate-Stabilized Total Knee Arthroplasty Reproduces Native Medial Collateral Ligament Strain"

_medicina, 2022, doi:10.3390/medicina58121751_

Round 1

Reviewer 1 Report (Previous Reviewer 2)

Author Response

Reviewer 1

Thank you for considering my concerns. In most cases, your amendments and complements are satisfactory. However, two major issues have not been addressed adequately (please see below).

  1. The authors mentioned in the text (p. 5, l. 142 146), that an ANOVA repeated measures was conducted. Considering overall eight cadaveric knees and making assumptions about the normal distribution is questionable. Have the authors conducted a test for normal distribution?

▶The authors are aware of the reviewer's concerns. We have assessed the normality of our data set using the Shapiro–Wilk test and confirmed that our data set was normally distributed. We added this issue to the statistical analyses section of the re-revised manuscript (Lines 140 - 141).

  1. Performing only three cycles and choosing one for statistical analysis is not comprehensible. How did the authors check repeatability? Why not calculate the average of the three cycles? I presume that the authors had concerns in terms of muscle fixations and possible loosening. (?) Admittedly, even three cycles in such measurements seem to be too little. Based on my experience, particularly the first and the last cycles of a cycle sequence are usually a little bit off before showing constant curve progressions. Please comment on that issue. Preferably, the authors could provide exemplarily the corresponding time series of the three flexion-extension cycles (not in the manuscript but in the response section).

▶The authors recognize the reviewer’s concern regarding repeatability and agree with the excellent point. Before the real assessment, we repeated flexion-extension of the cadaveric knee when potting the specimen in a cylindrical mold in resin to achieve more smooth flexion-extension motion. In addition, during the actual measurements, we tried to raise reliability for repeatability by comparing and confirming the repeatability in a real-time fashion. After the three flexion-extension cycles, all of the graphs were plotted at the same time and analyzed using with the SPM 1D package MATLAB version (https://spm1d.org). In every assessment, all of the three graphs were closely matched and there were no group-differences with statistical significance. We added this issue to the Method section of the re-revised manuscript. (Lines 120-122). In addition, as requested, we provided a corresponding time series of the three flexion-extension cycles below. However, unfortunately, we only kept the repeatability test data of another assessment on file. This test was performed before every assessment of MCL strain. It assessed the medial and lateral rollback between preoperative and post-TKA knees using the motion capture analyses system and optical markers. The red lines indicated preoperative medial rollback and the blue line indicated preoperative lateral rollback. The pink line indicated post-TKA medial rollback and the green line post-TKA lateral rollback. Each group of three graphs followed the same color coordination, and each graph was plotted after every single cycle. As you can see, all three graphs of each group were closely matched. We hope that these revisions address the reviewer’s concerns satisfactorily.

Reviewer 2 Report (Previous Reviewer 1)

This  reported nearly normal strain of MCL after BCS TKA in cadaveric study. These findings may be a tool to clarify  complaint of medial knee pain after TKA.

Author Response

Reviewer 2

This reported nearly normal strain of MCL after BCS TKA in cadaveric study. These findings may be a tool to clarify complaint of medial knee pain after TKA.

▶The authors appreciate the reviewer’s succinct summary of our study. We believe that no changes to the text are necessary.

Round 2

Reviewer 1 Report (Previous Reviewer 2)

This manuscript is a resubmission of an earlier submission. The following is a list of the peer review reports and author responses from that submission.

Round 1

Reviewer 1 Report

This manuscript revealed a small MCL strain in BCS TKA  compared normal knee in cadaveric  study. I have some concerns about video extensiometer analyses as mentioned in the discussion. Is there any paper reported the accuracy of the equipment ? Or did the Authors confirmed the accuracy ? And  the same  knees were compared before TKA and after TKA. Is there some difference before and after, such as tight MCL before TKA and loose MCL  after TKA ? In the clinical course of the TKA, some repairing will be occur.

Reviewer 2 Report

Dear authors,

Thank you very much for submitting your interesting and meaningful work. Overall a soundly conducted study is presented. However, several passages need to be revised (please see below).

1.       The statement "These findings provide clues to understand why patients undergoing guided-motion BCS TKA may experience more natural feeling knee, compared to conventional TKA" in the conclusion section of the abstract is too speculative.

INTRODUCTION

2.       Please paraphrase the statement "replicate native knee kinematics" (p. 2, first paragraph). I am unaware that there are knee endoprostheses that can replicate native knee kinematics. TKAs, such as the BCS TKA, reveal similarities, but they still differ from native kinematics, particularly rotations around the longitudinal axis (DOI:https://doi.org/10.1016/j.arth.2017.09.035). 

3.       p. 2: "In ACL-deficient conventional TKA design, the MCL and remaining capsular structures provide restraint against anterior tibial translation. Previous studies have reported that MCL strain was worse after conventional ACL-deficient TKA than in native knees [12-16]. Theoretically, BCS TKA may restore post-TKA MCL strain to levels observed in native knees by enhancing anterior-posterior stability."

The authors provided a reasonable explanation.

4.       The study purpose and hypothesis are clearly stated.

METHODS

5.       In terms of physiological muscle loading, the used muscle forces are too small to simulate even level walking in a normal-weight person (p. 3). Based on my experience, I know very well how hard it is to apply realistic forces in a human specimen. I think that this is a major methodological issue in this study. The authors would have to validly extrapolate the measured values to rather physiological loading conditions, which is not an easy task. Please comment on that issue. 

6.       Please provide more details about the utilized squatting simulator (i. a., angular velocity, type of actuators, closed or opened loop control system …). The authors mentioned the "original Oxford rig". Please provide, if possible, a respective reference that potentially indicates the specifications of the simulator.

7.       According to the respective figures, the authors conducted multiple t-tests. Conducting multiple t-tests increases the likelihood of type 1 error considerably. Please utilize other statistical methods enabling an analysis considering the entire time series (e.g., statistical nonparametric mapping, PCA). The current approach is not valid.

8.       How many flexion-extension cycles were used for statistical analysis?

RESULTS

9.       Please provide mean, standard deviation and percentage values when referring to your results.

DISCUSSION

10.   How do the authors explain the high standard deviations in Figures 5B and 6B(M3)?

11.   The passage (p.8, first paragraph) "Our findings, when taken into account with this anatomical circumstance, suggest that guided-motion BCS TKA may lead to reduced postoperative pain, …" seems speculative. Are there studies that support this hypothesis?

12.   Generally, the authors' explanations regarding the BCS TKA kinematic benefits seem to be a little bit too subjective. Since the authors did not acquire 3D kinematics, drawing inferences about the results' causality is not valid.